# The Role of Lipids in the Process of Hair Ageing

**Luisa Coderch [1], Ritamaria Di Lorenzo [2] , Marika Mussone [2], Cristina Alonso [1] and Meritxell Martí [1,*]**

[1] Surfactants and Nanobiotechnology Department, Institute of Advanced Chemistry of Catalonia (IQAC-CSIC), Jordi Girona 18-26, 08034 Barcelona, Spain

[2] Department of Pharmacy, University of Naples Federico II, Naples, Via Domenico Montesano 49, 80131 Napoli, Italy

[*] Correspondence: meritxell.marti@iqac.csic.es; Tel.: +34-932557644

**Abstract:** An obvious sign of ageing is the loss of hair colour due to a decrease or lack of melanin in hair fibres. An examination of the lipid levels and structure of grey hair determined using μ–FTIR revealed a high correlation between the characteristics of lipids located in the cuticle and the water dynamics of the fibres. Therefore, a deep study based on external and internal lipid extraction, an analysis using thin layer chromatography coupled to an automated flame ionisation detector, calorimetric analyses and the physico-chemical evaluation of the delipidated fibres were performed. Hairs were evaluated to identify changes in the organisation of these lipids using Fourier transform infrared spectroscopy and their effect on the water dynamics of the fibres. The primary differences observed for the lipid extracts from white hair compared to brown hair were the lower amount of the internal lipids extracted, which were primarily composed of free fatty acids (FFAs) and ceramides, with a higher content of lower phase transition peaks, indicating increased unsaturated compounds that promote higher fluidity of the lipid bilayers. The virgin white fibres exhibited lower levels of embedded water, with lower binding energies and higher water diffusion, indicating higher permeability. The IR study confirmed the low lipid levels and the greater disorder of white hair. These results may be of interest for cosmetic treatments to which patients with grey hair may be subjected.

**Keywords:** white hair; ageing; hair treatment; lipids; spectroscopy ATR-FTIR; calorimetric analysis

## 1. Introduction

Hair ageing comprises weathering of the hair shaft [1] and ageing of the hair cuticle [2]. The hair is composed of two distinct anatomical structures: the hair shaft, which is the visible part above the surface of the skin and the hair follicle, which represents the subcutaneous portion.

The hair follicle is the site of pigmentation within the hair. In hair follicles, melanocytes synthesise melanin within melanosomes [3]. In the dermal papilla, melanosomes are transferred to precorneal keratinocytes, which differentiate and organise to form the pigmented hair shaft [4,5].

The hair shaft is a filament-like structure derived from the epidermis that arises from the hair follicle. This complex biological material is composed of three concentric layers. The cuticle is the most external layer and is formed by planar cells that overlap the rest of the fibres [6]. The cuticle forms a protective barrier reinforced by a lipid network, accounting for only 1–9% of the hair bulk but serving a crucial function in maintaining the hydrophilicity of the hair and protecting it [7]. The cortex, which accounts for most of the fibre mass, is made up of elongated keratin inserted in a protein matrix. It plays an indispensable role in providing resistance and flexibility to the fibre [8], and it contains melanin in the pigmented hairs [5]. Finally, the medulla, which is not always present, is located in the innermost part of the hair. Recent studies confirmed the previous detection of lipids in this region [9,10]; in fact, lipid concentrations were found to be 3–20-fold higher in the medulla than in the surrounding cortex [11,12].

Greying has been largely investigated at the hair follicle level. Several studies support the weakening of the antioxidant system and progressive accumulation of ROS in grey hair follicles [13–15], resulting in the oxidation of amino acids, such as methionine and affecting the activity of the enzymes involved in melanogenesis [16–18]. The hair's lipid composition may also be altered during age-related colour loss along with a decrease in the melanin content, which may result in an alteration of the fibre characteristics. To date, studies on hair lipids have mainly been performed using the comparison of hair from diverse ethnic groups or by hair exposure to external conditions [19,20]; very few studies have focused on the underlying processes of pigment loss [21]. Despite the importance of lipids to hair properties, few studies have profiled the lipids present in both pigmented and unpigmented fibres. The reduced exogenous and endogenous lipid profile of grey hair includes lower levels of 18-MEA and decreased de novo synthesis of dihydroceramide [22–24]. One study found that hair cholesterol increases with age in women only [25]. The differences in the lipid concentration or the fluidity of lipids that are involved in the barrier function in hair fibres could account for the noticeable physical characteristics of pigmented and nonpigmented hair fibres [26]. Moreover, recent works have highlighted the importance of lipid modification in greying hair. Lipids in the hair follicle root were found to differ between brown and white hair. Phospholipids, vitamin D3 and cholesterol are significantly decreased in white hair follicles [27]. The reduced lipid content of all hair shaft sections—the cuticle, cortex, and medulla—occurs with advancing age and affects the clinical properties of grey hair [11]. The lipid order structure was also found to be modified in greying hair [28], and a gradual modification of lipid conformations (trans/gauche ratio) was associated with changes in grey hair shafts related to the alteration of the barrier function [29].

Therefore, the primary objective of this work was to increase our knowledge of the role of exogenous and endogenous lipids in brown and white Caucasian hairs to determine their impact on selected physico-chemical properties. In particular, the solid/fluid status of lipids and the arrangement of lipid bilayers are associated with properties such as water uptake, desorption and permeability kinetics. In this study, the lipid content of brown and white Caucasian hairs was characterised. Both the sebum and free structural lipids were removed from the surface and from within the fibre in our extraction process. Thus, it is possible to discriminate between differences in the functional/structural contributions of sebaceous or free structural lipids to the fibre properties measured from pigmented and unpigmented hairs.

## 2. Materials and Methods

Brown and white Caucasian hairs belonging to different people or the same person were externally and internally subjected to lipid extraction and analysed using thin-film chromatography coupled to a flame detector (TLC-FID). A calorimetric study of the different lipid extracts was performed using thermogravimetric analysis (TGA) and differential scanning calorimetry (DSC). In addition, virgin and lipid-depleted fibres of pigmented and nonpigmented fibres were evaluated to discern the influence of lipids in both fibres. Dynamic vapour sorption (DVS) was performed to determine the permeation characteristics of both hair fibre types. Differences in the lipid order or quantity in both types of hair were examined on the surface using ATR-FTIR and at the different layers of the fibre using μ-FTIR. Both techniques were chosen for their complementary approaches. The combination of DVS and FTIR makes it possible to describe the interactions of the fibres with the environment as well as the amount and structure of the lipids in the region that regulate this change.

### 2.1. Hair

The hair samples used in this study were natural virgin brown Caucasian and white Caucasian hairs, both without any pretreatment, which were provided by De Meo Brothers (Passaic, NJ, USA). In addition, brown and white Caucasian hairs from the same volunteer were also studied (49-year-old female volunteer (Catalonia, Spain)). All hair samples

were washed with 3% diluted Pantene Pro-V commercial shampoo (Procter & Gamble, Cincinnati, OH, USA) at a hair/surfactant solution 1/30 bath ratio, followed by a water rinse and drying under ambient conditions.

Shampoo Pantene ProV contains the following compounds: water, sodium lauryl sulfate, sodium laureth sulfate, cocamidopropyl betaine, sodium citrate, sodium xylene-sulfonate, fragrance, sodium chloride, citric acid, sodium benzoate, hydroxypropyl methyl-cellulose, tetrasodium EDTA, panthenol, panthenyl ethyl ether, methylchloroisothiazoli-none and methylisothiazolinone.

### 2.2. Lipid Extraction

Extractions of external and internal lipids were performed using different organic solvents. The external surface lipids were initially removed from the washed hair surface using Soxhlet extraction with t-butanol and n-hexane for 4 h [20]. Internal lipids were extracted using different solvent mixtures of chloroform/methanol (2:1 $v/v$, 1:1 $v/v$ and 1:2 $v/v$), with each mixture applied for 2 h. Then, 100% methanol was applied to the hair samples overnight by stirring at room temperature. The individual extracts were then pooled, concentrated and dissolved in a chloroform/methanol mixture (2:1 $v/v$) before analysis [7,20].

### 2.3. Lipid Analyses

Lipid analyses of the individual samples were conducted using thin-layer chromatography coupled to an automated flame ionisation detector (TLC/FID) (Iatroscan MK-5, Iatron, Tokyo, Japan). Lipid samples were directly stained on chromarods coated with silica gel (type S-III) using a 2 μL precise Hamilton syringe connected to an SES 3202/IS-02 sample spotter (NiederOLm, Germany). Lipid determination was performed using an optimised TLC/FID procedure [7,20] in which the rods were run in mobile phases as follows: (i) chloroform/methanol/water (57:12:0, $v/v/v/v$), (ii) chloroform/methanol/water (57:12:0.6, $v/v/v/v$) over a distance of 2.5 cm twice, (iii) n-hexane/ethyl ether/formic acid (50:20:0.3, $v/v/v/v$) at 8 cm and (iv) n-hexane/benzene (35:35, $v/v/v$) at 10 cm. Finally, a total scan (100%) was conducted to quantify the major polar lipids.

### 2.4. Calorimetric Analysis

Differential scanning calorimetry (DSC) was performed using a Mettler Toledo DSC Model DSC-281 equipped with a Dewar containing liquid nitrogen (IQAC-CSIC Thermal Analysis and Calorimetric Service "Josep Carilla", Barcelona, Spain). The lipid extract samples (5–10 mg) were placed in a hermetically sealed aluminium pan, cooled under a stream of nitrogen from 25 to $-100$ °C at a cooling rate of 10 °C/min and immediately heated from $-100$ to 100 °C at a heating rate of 10 °C/min.

Thermogravimetric analyses were performed using a TGA instrument (Model TGA/SDTA 851; Mettler Toledo, Barcelona, Spain) (IQAC-CSIC Thermal Analysis and Calorimetric Service "Josep Carilla", Barcelona, Spain). Approximately 5 mg of extract was packed into a pierced aluminium pan (100 μL) and heated under a nitrogen stream from 25 to 550 °C at a heating rate of 10 °C/min.

### 2.5. Moisture Content

Hair samples of nonextracted, external lipid-extracted and internal lipid-extracted hair were evaluated for moisture content. A 0.5 g hair sample was kept in a conditioned room (23 °C, 50 % RH) for at least 24 h before being weighed and then dried in an oven at 105 °C for 12 h. The hair sample was dried in a desiccator under a $P_2O_5$ atmosphere. After cooling in a desiccator under $P_2O_5$, the sample was weighed, and the moisture content (%) was calculated.

*2.6. Dynamic Water Vapour Sorption*

A Sorption Analyser Q5000SA thermogravimetric balance (TA Instruments, New Castle, DE, USA) with a controlled humidity chamber was used to determine water absorption and desorption in the hair fibres (IQAC-CSIC Thermal Analysis and Calorimetric Service "Josep Carilla", Barcelona, Spain). The experiments were performed on each hair sample in triplicate (10 ± 1 mg) with a total gas flow of 200 mL/min at 25 °C according to a protocol described elsewhere [20,30]. To obtain the diffusion coefficient, the method applied by Vickerstaff [31] was used. This is expressed using an equation derived from Fick's equation applied to the diffusion of moisture. With this expression, satisfactory results were obtained for the first stages of moisture absorption. The fraction of absorbed water represented versus the square root of the absorption time should fall on a linear line whose slope is the square root of the apparent diffusion coefficient, Da. The coefficient of apparent diffusion is expressed in min $^{-1}$ over the mass of the sample.

*2.7. Infrared Analyses Using FTIR*

A Nicolet Avatar 360-FTIR spectrophotometer (Madison, Wisconsin, USA) equipped with an attenuated total reflection (ATR) accessory including a diamond crystal (with ZnSe lens) with a 42° angle of incidence in horizontal orientation was used. Before analysis, the hair was placed towards the diamond crystal. To ensure reproducible contact between the sample and the glass, a pressure of 10,000 psi was applied to the samples. All analysed spectra represent an average of 64 scans with a resolution of 2 cm$^{-1}$; the wavenumber range used was 4200–650 cm$^{-1}$. The maximum positions were determined using OMNIC software version 8.1.210 (ThermoFisher, Sci, Waltham, MA, USA).

The μ-FTIR was performed with a MIRAS beamline at the ALBA Synchrotron 35 using a Hyperion 3000 Microscope (Bruker) equipped with a 36× magnification objective and condenser coupled to a Vertex 70 spectrometer (Bruker) with a 50 μm HgCdTe (MCT) detector that was continuously purged with N2 gas, as described elsewhere [11,32,33]. Hair tufts (1 cm) were embedded into an optimal cutting temperature (OCT) compound (Bright Instruments) and immediately frozen using liquid N2. The sample blocks were cut into 5 μm cross-sections using a Cryostat CM3050 S (Leica Biosystems, Nussloch, Germany). Sections were placed on 13 mm CaF2 circular windows with a 1 mm thickness. The measuring range was 4000–900 cm$^{-1}$, and the spectra collection was performed in a transmission mode of 4 cm$^{-1}$ resolution, 10 μm × 10 μm aperture dimensions, a step size of 10 μm and 128 coadded scans. Spectra were corrected using the baseline offset, and a linear baseline correction was provided by the software.

## 3. Results and Discussion

The lipid differences between brown and white Caucasian hairs were found in a previous study of the structure, and the amount of lipids in the different parts of the fibre were found using μ–FTIR mapping, and their relationship was found with the barrier function of both fibres [11]. Primarily, the cuticle of the white hair fibre exhibited a significant decrease in lipid content but did not display much difference in lateral packing order compared to the cuticle of the brown hair fibre. The DVS analyses indicated a decreased capacity in white hair to absorb water with an increase in the velocity of the exchange of water with the environment. The lower capacity to absorb water may be more related to the protein differences between fibres; however, the decreased levels of lipids could be more related to the increase in the water diffusion of the white fibres. Therefore, the amount and type of lipids in the different fibres should be determined using different techniques, and the effect of delipidation should be determined to support the effect of the lipids in the transition from brown to white.

*3.1. Lipid Extraction*

The analyses of the lipid extracts might provide interesting knowledge regarding the levels and types of lipids that can be obtained from the two kinds of hair studied.

Lipid extraction from white and brown Caucasian hair was performed after washing with shampoo. As detailed in the Materials and Methods Section, the external surface lipids were extracted from the hair surface using Soxhlet extraction with t-butanol and n-hexane; the internal lipids were extracted with different chloroform/methanol mixtures. This methodology was performed in duplicate. The results of the lipid levels extracted and the different lipid families determined using the TLC/FID analyses expressed in weight percentage over the whole weight of fibre are shown in Table 1.

**Table 1.** Total amount of lipids and different lipid families obtained using extraction with tert-butanol/hexane (external lipids (E)) and chloroform/methanol mixtures (internal lipids (I)) as a percentage of hair mass of brown (B) and white (W) Caucasian hair fibres.

|  | BE | WE | BI | WI |
|---|---|---|---|---|
| **Total Lip Ext (% owf)** | **0.92 ± 0.11** | **0.81 ± 0.15** | **2.48 ± 0.39** | **1.52 ± 0.16** |
| Apol Lip (% owf) | 0.13 ± 0.07 | 0.10 ± 0.02 | 0.20 ± 0.07 | 0.11 ± 0.04 |
| FFA (% owf) | 0.28 ± 0.08 | 0.34 ± 0.06 | 0.98 ± 0.15 | 0.43 ± 0.15 |
| Sterols (% owf) | 0.04 ± 0.02 | 0.01 ± 0.00 | 0.12 ± 0.05 | 0.08 ± 0.01 |
| Polar Lip (% owf) | 0.08 ± 0.06 | 0.08 ± 0.02 | 0.32 ± 0.06 | 0.23 ± 0.08 |

External lipid extracts for the two fibres accounted for similar total levels, which were approximately 1%. Each lipid family was also present in a similar proportion, with the highest amount for FFA at approximately 0.3% and the lowest amount for sterols at approximately 0.02%. Of note, the fibres were previously washed with shampoo, which does not remove all free lipids from the fibres' surface as an essential amount of lipids remains on the surface layers.

The internal lipid extracts account for higher levels than the external extracts at approximately 1.5% for white hair and 2.5% for brown hair. This indicates that grey or white hairs contain decreased levels of internal lipids, which is in accordance with the previously published IR results [11]. The TLC/FID analyses demonstrated a greater amount of FFA and polar lipids (ceramides) in the internal lipid extract of the brown hair than in the white hair. Other investigators reported a lipid decrease in the unpigmented hair. Wang et al. found a significant decrease in the phospholipid, cholesterol and vitamin D3 contents in the white hair follicles [27]. Additionally, a modification of the ceramide content in unpigmented hair has been previously reported [24].

Therefore, it can be concluded that there are significant differences in the total levels and types of lipids extracted from brown and white hairs, which indicates a loss of lipids, primarily FFA and polar lipids, in the brown-to-white transition. However, the two types of hair used for this study are commercial hair of different origins, so these lipid modifications could be due to their different origins rather than from the transition to grey hair. For this reason, a parallel study was performed using the same extractions on hair from a single volunteer who had both brown and grey hair.

Figure 1 compares the external and internal lipid extracts of brown and white hair from the commercial source and single volunteer. The data are expressed as the percentage of total extract analysed to determine the differences between them more clearly.

It is important to emphasise the very similar profile obtained from the extracted lipids derived from the commercial hair compared to the volunteer hair. All extracts exhibited a predominant amount of FFA. In addition, the greater amount of FFA was more apparent in the white hair for external extracts and in the brown hair for internal extracts. Polar lipids were dominant in the external extracts; sterols and polar lipids were much more prevalent in the internal extracts, being primarily more enriched for ceramides in the white hair (the primary component of polar lipids).

The similar lipid profiles obtained from the commercial white hair and volunteer compared to the brown hair support the primary conclusions of significantly reduced levels of lipids in white hair, mainly due to a decrease in the FFA family. These results support further evaluations in the present study using commercial hair samples.

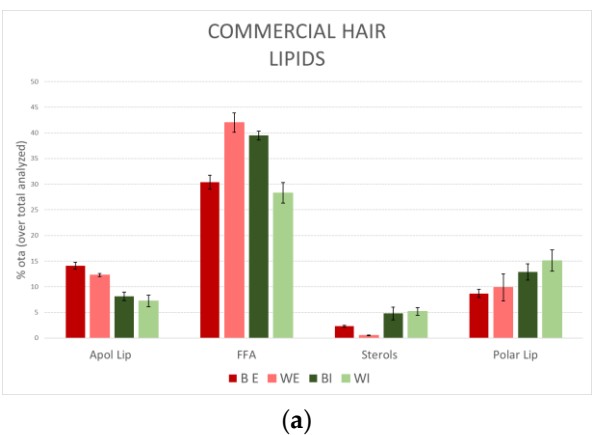
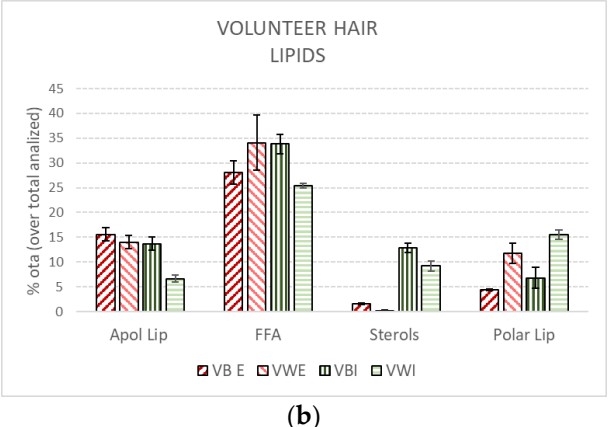

(**a**)                                                 (**b**)

**Figure 1.** Percentage of lipid families obtained using extraction of brown (B) and white (W) Caucasian hair with tert-butanol/hexane (external lipids (E)) and chloroform/methanol mixtures (internal lipids (I)) from a commercial source (**a**) and from a single volunteer (**b**).

### 3.2. Thermogravimetric Analysis of Extracted Lipids

The analyses of lipid extracts using DSC and TGA allow for a better understanding of the behaviour of lipids as a whole. The DSC results indicate that the four extracts have two primary phase transitions: a main peak ($-17$ to $-37$ J/g) at a low temperature (10 to 20 °C) and a second peak ($-2$ to 12 J/g) at a slightly higher temperature (30–44 °C). Comparing the brown and white extracts (Table 2), the low-temperature peak was more prominent in the white lipid extracts, and the high-temperature peak was more evident in the brown extracts, indicating that most fluid lipids were extracted from the white hair. Of note, the increase in the phase temperature of almost 10 °C for the white hairs occurred primarily in the case of the external extracts, whereas more similar thermotropic behaviour was obtained for the internal extracts. Therefore, even with the small increase in peak temperatures, particularly in the external lipid extract from the white hair, the highest enthalpy of the high-temperature peaks of the lipid extracts of brown hair indicates that the brown lipid extracts have more saturated lipids with higher transition temperatures.

**Table 2.** Results of thermogravimetric analyses of external (E) and internal (I) lipid extracts of brown (B) and white (W) hairs.

|  | BE | | WE | | BI | | WI | |
|---|---|---|---|---|---|---|---|---|
| **DSC** | °C | J/g | °C | J/g | °C | J/g | °C | J/g |
|  | 10.5 | −24.7 | 19.4 | −37.3 | 12 | −16.7 | 17.5 | −26.2 |
|  | 30.3 | −11.5 | 41.7 | −4.4 | 43.8 | −4.6 | 38.0 | −2.5 |
|  | 54.9 | −6.4 | — | — | — | — | 50.2 | −0.8 |
| **TGA** | °C | % | °C | % | °C | % | °C | % |
|  | 242.8 | 36.1 | 247.1 | 34.1 | 249.0 | 64.6 | 241.3 | 63.0 |
|  | — | — | — | — | 340.6 | 25.6 | 338.2 | 19.1 |
|  | 457.8 | 23.9 | 463.0 | 26.3 | 461.5 | 3.3 | 445.2 | 5.5 |

— peak not found in this type of lipids.

Thermogravimetry is an effective method for elucidating the probable mechanism of solid-state reactions such as thermal decomposition and dehydration. The first-derivative thermogravimetric analysis (TGA) evaluation of the four extracts presented two or three main degradation steps depending on the extract (Table 2). The TGA was very different between the external and internal extracts but was not very different between all of the brown and white samples. The most important peak from 240 to 250 °C accounted for approximately 35% of the external extracts and 65% of the internal extracts. A second peak at approximately 340 °C accounted for approximately 20% of the internal extracts, and a

third peak, from 445 to 465 °C, accounted for 25% of the external extracts and only 5% of the internal extracts. Using this methodology, the temperature of decomposition is superior here at approximately 5 °C for the white hairs in the case of external extracts. However, the three peaks obtained for the internal lipid extracts showed a reduction in the temperature for the white hairs relative to the brown hairs. Ceramides in hair have been reported to predominantly contain non-hydroxy or α-hydroxy FFA and dihydrosphingosine (NDS) moieties [34,35], which are normally not present in the stratum corneum. In addition, increased saturated FFAs containing NDs were observed in the brown hair compared to the white hair [24]. This supports the low temperature of all peaks found for the internal lipid extracts, which are rich in ceramides, in white hairs. However, these results all indicate many differences between the external and internal extracts but not as many between the brown and white hair lipid extracts.

### 3.3. Hair Moisture

Hair properties are known to change in a significant and reproducible manner as a function of the relative humidity of the environment. Hair is able to absorb considerable quantities of water, almost 25% at high RH, which leads to the swelling of fibres and plasticisation of the structure. As swelling occurs, the outer protective cuticle scales are not held as tightly to the hair shaft; together with an overall increase in the volume of hair on the head, this leads to increased friction between fibres and a subsequent decrease in manageability. At the same time, the plasticising effect of water alters the mechanical properties. This means that the water content of hair is a central issue that can modify intrinsic hair properties [36].

Therefore, the moisture content was determined gravimetrically, and water absorption and desorption were assessed using DVS. The water content of virgin hair fibres and extracted fibres was obtained gravimetrically from the different hair shafts equilibrated in a conditioned room at 23 °C and 50% RH (Table 3).

**Table 3.** Gravimetrically moisture at 50%RH and DVS parameters of virgin Brown (B) and White (W) Caucasian hairs, external extracted Brown (BE) and White (WE) Caucasian hairs and internal extracted Brown (BI) and White (WI) Caucasian hairs.

|  | B | BE | BI | W | WE | WI |
|---|---|---|---|---|---|---|
| Moisture at 50% RH Grav | 9.61 | 9.97 | 10.99 | 8.13 | 9.83 | 10.76 |
| Regain at 95% RH (%) | 26.25 | 25.61 | 26.18 | 24.85 | 24.15 | 25.50 |
| Wm (%) | 0.0772 | 0.0732 | 0.0795 | 0.0789 | 0.0729 | 0.0793 |
| Cg | 5.915 | 6.548 | 5.667 | 5.813 | 6.460 | 5.455 |
| K | 0.7367 | 0.7480 | 0.7334 | 0.7209 | 0.7396 | 0.7260 |
| $R^2$ | 0.9980 | 0.9977 | 0.9989 | 24.85 | 24.15 | 25.50 |
|  |  |  |  |  |  |  |
| Da absor (min$^{-1}$ × 10$^{-3}$) | 16.41 | 17.41 | 14.85 | 17.27 | 18.14 | 16.57 |
| Da desor 95% (min$^{-1}$ × 10$^{-3}$) | 23.67 | 24.12 | 22.55 | 24.13 | 24.22 | 23.84 |
| Da (min$^{-1}$ × 10$^{-3}$) | 20.46 | 21.17 | 19.05 | 21.04 | 21.52 | 20.53 |

The virgin white hair exhibited a much lower moisture content than the brown Caucasian hair, with a mean value of 8.1% for white hair and 9.6% for brown hair. The external lipid extraction did not greatly affect the moisture content of the brown hairs but clearly increased the moisture of the white hairs to almost the same level as the brown Caucasian hairs, approximately 9.9%. Moreover, the internal lipid extraction induced an extra increase in moisture to almost 11.0% for both kinds of hair.

In accordance with the previous results [11], a higher moisture content was observed in the brown Caucasian hairs than in the white Caucasian hairs. In addition, the total lipid extraction of ethnic hairs was reported to induce an increase in the water regain of the hair fibres [30]. In our study, the external lipid extraction primarily affects the lipid structure of the white fibres, facilitating water permeation into the fibres to a level similar to that of

the brown Caucasian fibres. Moreover, the internal lipid extraction seemed to affect both kinds of fibres in a similar way. It seems that reduced lipids and decreased structural orders facilitate water penetration into the fibres.

A DVS analysis of the hair samples allowed us to understand the dynamics of the water absorption/desorption of the fibres. It has been used to study the moisturising efficacy of cosmetic products [37]. The DVS of the different samples indicates differences in the humidity content, the binding energies of water to different components of the fibre (GAB model) and the water diffusion or velocity to exchange water with the environment (Da), which is of special relevance in this study because of the important role of lipids in water diffusion (Table 3).

Water regain reveals the capacity of the fibre to uptake water and the humidity content at 95% relative humidity (regain at 95% RH), and it represents the maximal water that the fibre can acquire. The comparison between the two virgin fibres confirmed that the pigmented hair absorbed more water at higher levels of humidity (26.2% for brown hair versus 24.9% for white hair). This is in accordance with the moisture results at 50% RH that were determined gravimetrically. However, for extracted hairs, an increase in moisture was always detected at 50% RH. The results of regain at 95% RH indicated a small decrease in the water absorption of externally extracted fibres and a small increase in the water absorption of internally extracted fibres, primarily during the desorption process, resulting in similar values between the brown and white fibres. This can be clearly seen in Figure 2, in which there is a decrease in the amount of water for both the BE and WE external extracted fibres and a clear increase in the amount of water for the BI and WI internal extracted fibres, primarily in the desorption process. As previously stated [7], a decrease in the water content could be related to the modification of the external lipid barrier, which facilitates the loss of water content after external lipid depletion. Additionally, the extraction of internal lipids, which are primarily derived from the inner hair matrix cells, seems to facilitate water penetration, leading to more hydrated fibres. This accounts for both brown and white fibres presenting as the same, whereas the virgin and lipid-depleted white fibres exhibited reduced water content compared to the brown ones.

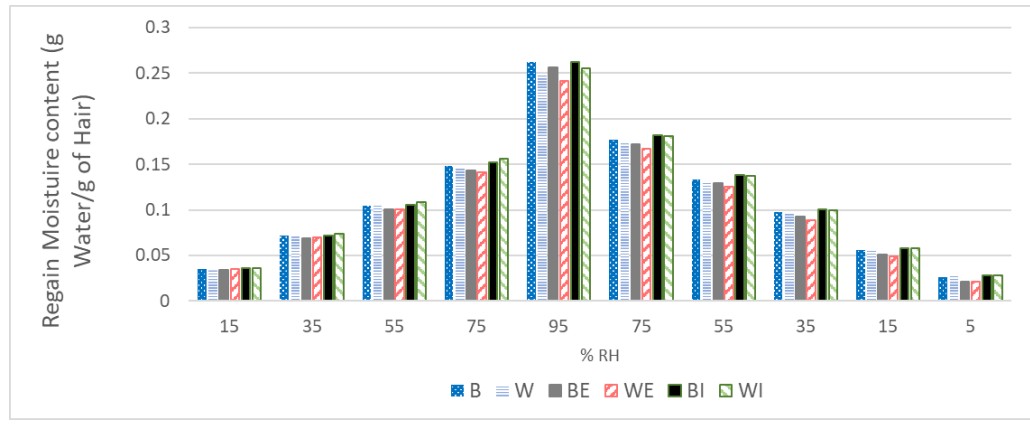

**Figure 2.** Water absorption isotherms of virgin brown (B) and white (W), externally extracted brown (BE) and white (WE), and internally extracted brown (BI) and white (WI) Caucasian hairs.

The GAB model yields values for different parameters: Wm is the monolayer capacity, and Cg and K are the energy constants [38] (Table 3). Similar to the maximum water regain, the water content in the monolayer decreased with external lipid extraction but increased after internal lipid extraction for both fibres. In addition, the energy constants Cg and K of the primary and secondary sorted monolayers, which were very similar for the two fibres, increasing after external lipid extraction and decreasing after internal lipid extraction, which led to values similar to those of the virgin fibres. These findings reveal an enhancement of the water binding energy to the active groups of the fibres following external lipid removal, which results in a more polar surface fibre. The internal lipid

extraction would make the inner part of the fibre more polar, which could allow an increase in water primarily in the protein fraction. Therefore, all results support decreased levels of water and lower binding energy for the white fibres relative to the brown fibres, although exhibiting similar behaviour when they are lipid-depleted.

An important parameter that may explain the water dynamics is the apparent diffusion coefficient (Da), which has been evaluated as a mean value of the absorption process (Da absorption), desorption process (Da desorption) and whole process (Da) (Table 3). The different Da values at certain humidities in the absorption and desorption process are shown in Figure 3. The kinetic evaluation of the absorption/desorption process is a good strategy for determining the structural integrity of the fibres. First, the diffusion of white hair was always superior to that of brown hair, independent of whether the samples were virgin or lipid-depleted fibres, as previously described [11]. Changes in the water content were also found to occur more rapidly in unpigmented hair [21]. Moreover, for the two fibres, there was an expected slight diffusion increase in the external lipid extracted fibres. However, diffusion surprisingly decreased after the internal lipid extraction, with a marked decrease for brown hair, from which more internal lipids were extracted. Similar results were obtained after the internal lipid depletion of Caucasian hair fibres in a previous work [7]. The diffusion of other lipid-depleted keratinised tissues, such as wool [39], nails [40] and stratum corneum from skin [41] has also been evaluated. Contrary to the results obtained here to a different extent, in all cases, the permeability of these tissues tended to increase after lipid extraction, indicating a partial loss of their integrity. Hair lipids were shown to exhibit a completely different repartition than those present in skin. Lipids seem to be deposited randomly in discrete nonkeratinised areas or as diffuse components throughout the material [42]. In a recent work, delipidated hair evaluated using AFM indicated that the morphological components of hair (i.e., cuticle lamellar structures and macrofibrils) become swollen [43], which could be the reason for the decrease in the diffusion. White hair and brown hair seem to be less permeable to water after total lipid extraction, indicating that the barrier is reinforced. Therefore, higher diffusion was obtained for all white fibres, with a similar behaviour when the lipids were depleted. This highlights the decrease in diffusion for the internal lipid-depleted fibres, which was more marked for the brown hair from which more internal lipids were extracted. Therefore, the white fibres are more permeable than the brown fibres in all cases. The influence of the lipid presence and extraction from brown and white fibres on permeation could help us understand the different chemical stresses imposed by straightening and colouring treatments applied to white fibres [23,26].

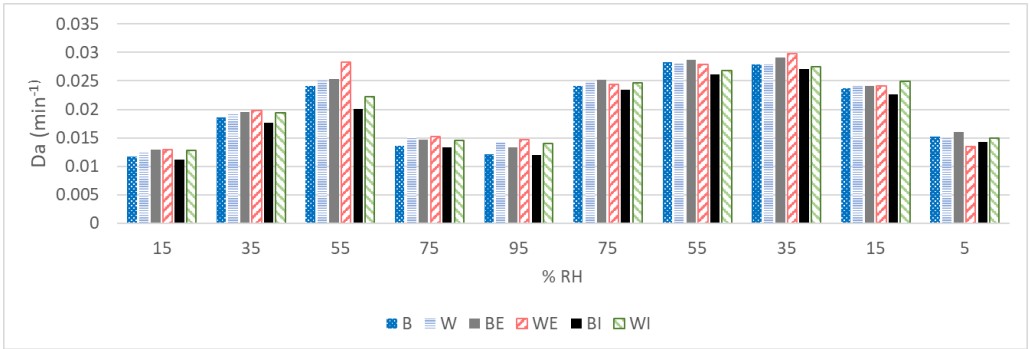

**Figure 3.** Apparent diffusion coefficients of virgin brown (B) and white (W), external extracted brown (BE) and white (WE), and internal extracted brown (BI) and white (WI) Caucasian hairs.

### 3.4. Hair IR

The possible differences between different hair fibres due to their different compositions and structural lipid organisation were evaluated using FTIR-ATR. This technique determines the lipid order disposition. For example, the $CH_2$ symmetric stretch mode position would indicate an orthorhombic, lamellar or gel lipid order with increased perme-

ability. The integration of the $CH_2$ asymmetric stretching mode is also quite reliable for determining lipid content.

The detection of vibrations in the alkyl chain within lipids and the resulting determination of the lipid conformational order and alkyl chain packing in both brown and white virgin and lipid-depleted Caucasian hair fibres was one of the primary objectives of this investigation. The levels and structural order of the lipid bilayers of the hair fibres account for their permeability properties and their water dynamics. To this end, an ATR-FTIR study was performed in which the surface of the virgin and lipid-depleted fibres was evaluated. Furthermore, a μ-FTIR study was performed on the virgin and lipid-depleted hair cross-sections to evaluate the three hair layers: the cuticle, cortex and medulla. This aimed to gain an increased understanding of the levels and the conformational order of the lipids of the different hairs and in the different hair layers.

The two main peaks in the IR spectra that reflect the phase of the lipids are $CH_2$ asymmetric stretching and $CH_2$ symmetric stretching peaks with maximums of approximately 2920 and 2850 $cm^{-1}$, respectively. In both peaks, a shift to higher wavenumbers indicated an increase in the lipid disorder. The 2850 $cm^{-1}$ peak was most susceptible to packing changes; a peak below 2850 $cm^{-1}$ is indicative of the presence of an orthorhombic (OR) chain lipid conformation, whereas a maximum between 2850–2852 $cm^{-1}$ is produced by a hexagonal (HEX) chain conformation, and a liquid crystalline (LIQ) chain conformation results in a maximum at a wavenumber higher than 2852 $cm^{-1}$ [44,45]. Shifts in the symmetric peak at 2820 $cm^{-1}$ are more imprecise but useful to corroborate the asymmetric peak at 2850 $cm^{-1}$.

The numerical values of frequency and intensity peak reduction after lipid depletion of the $CH_2$ stretching asymmetric and symmetric modes of both fibres in their virgin state and after external and internal lipid depletion are presented in Table 4. The frequencies of both stretching bands indicate the highest lipid order structures for the native brown hair compared to the white hair and both colours of lipid-depleted fibres. Lipid depletion tended to increase the frequencies in the two cases, which means increased disordered structures in the lipid-depleted fibres, with the difference being more pronounced in brown hair. Even though ATR-FTIR is not a fully quantitative technique, the comparison of peak amplitude or height can give us an idea of the variation in the lipid levels. Considering the smaller peak of the virgin white hair compared to the virgin brown hair (Figure 4), the lipid decreases for externally extracted fibres, accounting for approximately 20% and the internally extracted for 30%, being undetectable for WI.

**Table 4.** ATR-FTIR values of $CH_2$ stretching asymmetry and symmetric modes of virgin brown (B) and white (W), external extracted brown (BE) and white (WE), and internal extracted brown (BI) and white (WI) Caucasian hairs.

| | B | BE | BI | W | WE | WI |
|---|---|---|---|---|---|---|
| λ$CH_2$ St. Asym. $cm^{-1}$ | 2918.9 ± 0.1 | 2925.9 ± 0.3 | 2927.6 ± 0.2 | 2920.9 ± 0.6 | 2928.7 ± 2.4 | Undetectable |
| Intensity Reduction % | - | 18.5 | 23.9 | - | 16.6 | |
| λ$CH_2$ St. Sym. $cm^{-1}$ | 2850.1 ± 0.3 | 2851.5 ± 0.4 | 2851.7 ± 0.3 | 2851.0 ± 0.8 | 2851.0 ± 0.8 | Undetectable |
| Intensity Reduction % | - | 20.0 | 29.3 | - | 20.7 | |

Using ATR-FTIR, only the surface lipid structure and levels can be evaluated. However, lipid properties inside the fibre can be better examined using μ-FTIR analyses. To this end, 2850 $cm^{-1}$ was followed in the cross-sections of both hair types of the native and internal lipid-extracted hairs. The comparison of the frequency and height of the $CH_2$ symmetric stretching peak bands of the two types of virgin fibre confirmed a higher frequency value in white hair, especially in the cuticle and cortex. This indicates a greater lipid disorder in these regions of white hair and is consistent with the higher diffusion results for grey hair fibres. A recent confocal Raman microspectroscopy study identified a gradual modification of lipid conformation trans/gauche in greying hair for the first time [29]. The increase

in the gauche conformers obtained on unpigmented hairs supports the increase in lipid fluidity, as demonstrated by the peak frequency increase in the white hairs.

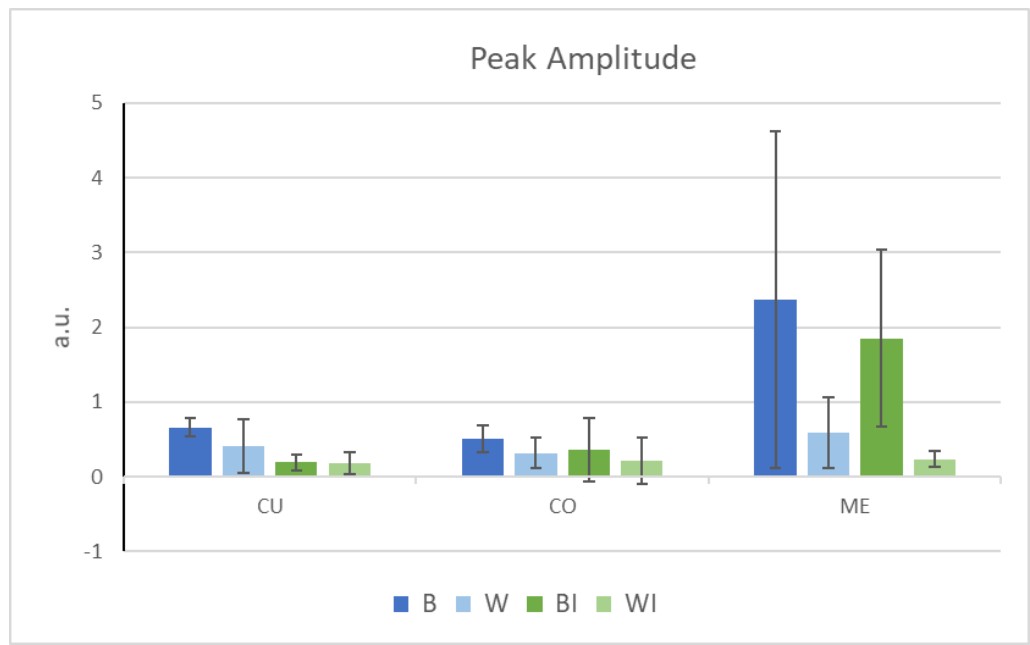

**Figure 4.** μ-FTIR peak amplitude of the CH$_2$ symmetrical stretching region of virgin brown (B) and white (W) (in blue) and internally extracted brown (BI) and white (WI) (in green) Caucasian hairs. (CU: cuticle, CO: cortex and ME: medulla).

Moreover, the peak amplitude (or peak height) was always lower in the case of the white fibres compared to the brown fibres in all three regions, with the difference being much more marked in the medulla (Figure 4). Thus, the levels of lipids were approximately 30% lower in the cuticle and cortex and 70% lower in the medulla in the white hair than in the brown hair. Previous work reported modification or a decrease in lipids in different hair layers. Lipids in the cuticles of grey hairs, especially the 18-MEA layer, have been reported to diminish, with uneven coverage of the surface [22]. Even with the known presence of lipids in the medulla [9,10], their chemical composition has only very recently been studied in pigmented hair [12]. A higher concentration of lipids in the medulla than in the cortex indicates great variability, ranging from barely noticeable to 20-fold higher [12]. They contain a mixture of nonesterified and esterified lipids, primarily saturated and carboxylate soaps, which could partially exhibit a crystalline phase. Recently, hair medulla was also found to contain globular structures assigned to the air-filled vacuoles interspersed with fibrillar ones, encompassed by a membrane, with the structural lipids providing the dominant interactions [46]. Squalene and oleic acid have also been detected [47]. Other authors found free fatty acids and wax esters [48], and a relationship was found between the unsaturated lipids of the medulla and glossines [47].

In accordance with the ATR-FTIR results of the fibre surface analyses, the total lipid extraction in the two fibres promoted a minimal modification in the cuticle at a higher frequency, indicating a tendency towards greater cuticular disorder in the two types of delipidated fibres. This increased disorder was much more marked for the grey fibres in both the cortex and medulla. However, the effect of delipidation on the brown fibres seemed to induce a trend of increased order in both the cortex and medulla (Figure 5). This greater structuring of the remaining internal lipids of the brown fibres after delipidation could explain the greater decrease in the aqueous diffusion observed in delipidated brown fibres.

Internal lipid delipidation creates similar lipid amount values for the brown and white fibres in the cuticle and cortex; however, there is a very little modification in the medulla of brown hair. This indicates much more difficult access for solvents to the medulla in

brown hair compared to white hair. Thus, the amount of lipids in the delipidated hairs is approximately only 10% in the cuticle, 40% in the cortex and from 20 to 60% in the medulla, with white hair always displaying fewer lipids.

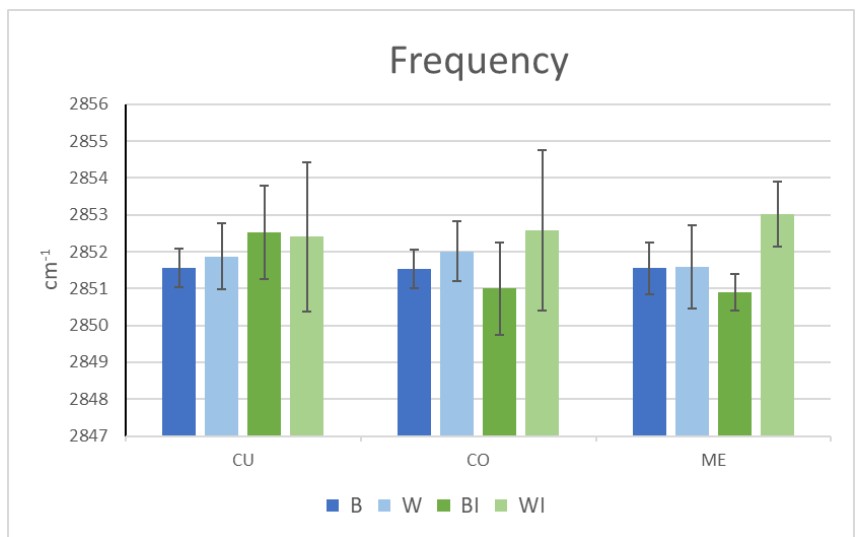

**Figure 5.** μ-FTIR peak frequency of the $CH_2$ symmetrical stretching region of virgin brown (B) and white (W) (in blue) and internally extracted brown (BI) and white (WI) (in green) Caucasian hairs. (CU: cuticle, CO: cortex and ME: medulla).

The FTIR results indicate that white hair exhibits lower lipid content than brown hair in all its layers, which diminishes after lipid depletion primarily in the cuticle and medulla. It is important to note the great variability of lipid content in the medulla and the elevated standard deviation, as previously reported [12]. In addition, the frequency was always higher for the white hair, which was more marked in the cortex and medulla, indicating greater unsaturation, increased fluidic layers and a marked disorder of the lipidic structures in these layers. The lower lipid levels and greater disorder, primarily at the cortex and medulla levels, support the greater diffusion of virgin and lipid-depleted white fibres, which are more important for internal lipid extraction. However, the lipid-depleted brown fibres showed an increase in the lipid order in the cortex and medulla, which can explain their decreased permeability results in the DVS study; this was also found elsewhere but is not easily explained [7,30]. The higher order of the lipid bilayer found in lipid-depleted brown fibres could hinder water penetration to a greater extent than in white fibres.

Greying has been largely investigated at the hair follicle level. Hair's lipid composition may also be altered during age-related colour loss at the same time as a decrease in melanin content, which may result in an alteration in the fibre characteristics. To date, studies on hair lipids have mainly compared hair from diverse ethnic groups or hair exposed to external conditions, but few studies have focused on the underlying processes of pigment loss [11,22,24]. However, recent works highlighted the importance of lipid modification in greying hair. Lipids in the hair follicle root differed between brown and white hair with a reduction in their content [27]; similar results were obtained in the different hair shaft sections: the cuticle, cortex and medulla [11]. In addition, a gradual modification of lipid conformation (trans/gauche ratio) was associated with changes in grey hair shafts related to the alteration of barrier function in greying hair [29]. Therefore, this work focused on the role of exogenous and endogenous lipids in brown and white Caucasian hairs to determine their influence on several physico-chemical properties. In particular, the fluid/solid-state of the lipids and the order of the lipid bilayers were related to properties such as water absorption, desorption and permeability kinetics.

External lipid extracts for the two types of fibres accounted for similar total levels of approximately 1%. Each lipid family was also present in a similar proportion, with the highest concentrations observed for FFA and the lowest concentrations for sterols. More

internal lipid extracts were found, approximately 1.5% in white hair and 2.5% in brown hair, with the highest levels of FFA and polar lipids (ceramides); these two lipid families were statistically greater for brown hair. The similar lipid profile obtained from commercial samples and a single volunteer with both brown and white hair lends support to the primary conclusion of significantly lower levels of lipids in white hair. The thermogravimetric study of external and internal lipid extracts from brown and white hairs was performed using DSC and TGA. This finding supports the marked differences between the external and internal extracts and the elevated unsaturation of the extracted lipids from white hair, probably due to the FFA and ceramide content.

Much lower moisture content was found for Caucasian white hair (8.1%) than for Caucasian brown hair (9.6%) (at 23 °C; 50% RH). There were decreased levels of water for both the BE and WE externally extracted fibres and a clear increase in the levels of water for the BI and WI internally extracted fibres, similar for both kinds of fibres, which indicates the different roles of external and internal lipids in the fibre. These results support lower levels of water and lower binding energy for the white fibres relative to the brown fibres and their similar behaviour when they are lipid-depleted. A kinetic evaluation of the absorption/desorption process is a good strategy for determining the structural integrity of the fibres. The diffusion of white hair was always superior to brown hair, independent of whether the samples were virgin or lipid-depleted, which indicates higher permeability in the white fibres. However, a decrease in the diffusion for the internally lipid-depleted fibres, which was more marked for the brown hair, could indicate a reinforcement of the lipid barrier.

The ATR-FTIR structural surface lipid study of the fibres indicated that the frequencies of both asymmetric and symmetric stretching bands presented the highest lipid order structures for native brown hair compared to white hair. Lipid depletion tended to increase the frequencies in the two cases, which means more disordered structures in the lipid-depleted fibres, with the difference being more pronounced for brown hair. A comparison of the peak amplitude or height revealed a marked lipid decrease in the external and internal extracted fibres. The μ-FTIR analyses of the lipid peaks inside the fibre confirmed a higher frequency value in the white hair, especially in the cuticle and cortex, indicating greater lipid disorder in these regions of the white hair. This is consistent with the higher diffusion results for the grey hair fibre, according to a reported gradual modification of the lipid conformation trans/gauche in greying [29]. Moreover, the peak amplitude was always lower in the white fibres in all three regions, with the difference being much more marked in the medulla. The total lipid extraction in both fibres promotes a minimal modification in the cuticle with an increase in disorder that was more apparent for the grey fibres in both the cortex and medulla. In contrast, the effect of delipidation on the brown fibres seemed to induce a trend of increased order in both the cortex and medulla. This greater structuring of the remaining internal lipids of the brown fibres after delipidation could explain the greater decrease in the aqueous diffusion of the delipidated brown fibres.

## 4. Conclusions

The primary differences observed for the lipid extracts from white hairs compared to brown hairs were the decreased levels of internal lipids extracted, which were primarily composed of FFAs and ceramides, with a higher content of lower phase transition peaks, indicating increased unsaturated compounds that promote higher fluidity of the lipid bilayers. In addition, the virgin white fibres exhibited lower levels of embedded water with lower binding energies and higher water diffusion, indicating higher permeability. The IR study confirmed the lower lipid levels and greater disorder of white hairs, especially in the cuticle. Total delipidation primarily modified the cortex and medulla, promoting disorder for white hair and order for brown hair.

This study supports reduced levels and a more disordered structure of the lipid bilayers in greying hair that promotes higher permeability compared to brown fibres. These results can be important for cosmetic treatments for grey-haired patients. Increased

permeability can increase the effects of common treatments such as perming, bleaching and dyeing. On the other hand, one could consider white hair repair formulations with the presence of fatty acids and ceramides with a high degree of saturation that are left on during the greying process.

**Author Contributions:** Data curation, M.M. (Marika Mussone) and C.A.; investigation, R.D.L.; writing—original draft, L.C.; writing—review and editing, M.M. (Meritxell Martí). All authors have read and agreed to the published version of the manuscript.

**Funding:** This research received no external funding.

**Institutional Review Board Statement:** Not applicable.

**Informed Consent Statement:** Not applicable.

**Data Availability Statement:** Not applicable.

**Acknowledgments:** Support for this work from CTQ2018-094014-B-100 is gratefully acknowledged. Moreover, the authors acknowledge the MIRAS beam-line at ALBA Synchrotron for the grant and the ALBA staff for their expert assistance. The authors also acknowledge the Service of Dermocosmetic Assessment (IQAC-CSIC) for their collaboration and technical support.

**Conflicts of Interest:** The authors declare no conflict of interest.

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
