# Peer review of "The Role of Lipids in the Process of Hair Ageing"

_cosmetics, doi:10.3390/cosmetics9060124_

Round 1

Reviewer 1 Report

This is interesting manuscript on a rising topic such as hair aging. 

recommendations for the authors:

1. improve title and abstract - add study design, setting, most important results

2. improve English language and scientific style

3. I find tables and figures appropriate but you could add few references from last three years on this matter 

4. Line 33 . Is missing 

5. Line 80 - this paragraph should be in the methods section 

Author Response

Comments and Suggestions for Authors Reviewer 1

This is interesting manuscript on a rising topic such as hair aging.

recommendations for the authors:

  1. improve title and abstract - add study design, setting, most important results

R: Abstract were completely changed following the reviewer's advices

Abstract: An obvious sign of aging is the loss of hair color due to a decrease or lack of melanin in the hair fibers. Examination of lipid levels and the structure of gray hair determined by µ–FTIR has revealed a high correlation between the characteristics of lipids located in the cuticle and the water dynamics of the fibers. Therefore, a deep study based on external and internal lipid extraction, analysis using thin layer chromatography coupled to an automated flame ionization detector, calorimetric analysis and physicochemical evaluation of the delipidized fibers were performed. Hairs were evaluated to identify changes in the organization of these lipids using Fourier transform infrared spectroscopy and its effect on the water dynamics of the fibers.

The primary differences observed for lipid extracts from white hair compared to brown hair were the lower amount of internal lipids extracted, which were primarily composed of FFAs and ceramides, with a higher content of lower phase transition peaks, indicating increased unsaturated compounds that promote higher fluidity of the lipid bilayers. Virgin white fibers exhibited lower levels of embedded water, with lower binding energies and higher water diffusion, indicating higher permeability. The IR study confirmed the low lipid levels and the greater disorder of white hair. These results may be of interest for cosmetic treatments to which patients with gray hair may be subjected.

2. improve English language and scientific style

R: The manuscript has been sent to its correction.

3. I find tables and figures appropriate but you could add few references from last three years on this matter 

R: The following recent references were added in the text with some comments.

43.McMullen, R.L; Zhang, G. Investigation of the internal structure of human hair with Atomic Force Microscopy J. Cosmet. Sci., 2020, 71,117-131

46.Fellows, A.P; Casford, M.T.L.; Davies, P.B. Using hybrid atomic force microscopy and infrared spectroscopy (AFM-IR) to identify chemical components of hair medulla on the nanoscale J. Microsc. 2021, 284, 189-202

47.Ymazaki, J.; Maeda, K Analysis of lipids in the medulla of Japanese hair and their function. Cosmetics2018 5(2), 27.

48.Kaneta, D.; Goto, M.; Hagihara, M.; Leproux, P.; Couderc, V.; Egawa, M.; Kano, H. Visualizing intra-medulla lipids in human hair using ultra-multiplex CARS, SHG, and THG microscopy. Analyst, 2021, 146(4), 1163-1168.

37.Kamath, YK, Quantification of human hair moisturization with cosmetic products by Dynamis Vapor Sorption J. Cosmet. Sci., 2020, 71, 303-320

18.O'Sullivan, J. D.; Nicu, C.; Picard, M.; Chéret, J.; Bedogni, B.; Tobin, D. J.; Paus, R.,The biology of human hair greying. Biological Reviews, 2021. 96(1), 107-128.

42.Bildstein, L.; Deniset-Besseau, A.; Pasini, I.; Mazilier, C.; Keuong, Y. W.; Dazzi, A.; Baghdadli, N. Discrete nanoscale distribution of hair lipids fails to provide humidity resistance. Analytical Chemistry, 2020. 92(17), 11498-11504.

4. Line 33 . Is missing 

R: Sorry but we do not find which is missing in line 33 

5. Line 80 - this paragraph should be in the methods section

R: According to the Reviewer, last paragraph of the Introduction, line 80, was moved to 2. Materials and Methods.

Reviewer 2 Report

Dear Authors,

After carful reading of your manuscript, its relevance and context in the field were clear, and it’s content could be a contribution to the hair care science. Initially, it is requested to authors to present the approval of the local ethics committee for your study, since a volunteer participated of the research design. Overall, distribute your references along the text, several paragraphs lacked them. Figures and tables missed legend. Figures lacked standard deviation bars. The conclusions were too long and it must exclusively answer the objectives. Some paragraphs from Materials and Methods could be more suitable in Discussion. It was not clear why a commercial shampoo was used, since it could add interferences to the analytical tests. Its composition must be described.

Author Response

Please find upload the answer to Reviewer 2
